# Biomimetically inspired asymmetric total synthesis of (+)-19-dehydroxyl arisandilactone A

Yi-Xin Han[1],*, Yan-Long Jiang[1],*, Yong Li[1], Hai-Xin Yu[1], Bing-Qi Tong[1], Zhe Niu[1], Shi-Jie Zhou[1], Song Liu[2], Yu Lan[2], Jia-Hua Chen[1] & Zhen Yang[1,3,4]

Complex natural products are a proven and rich source of disease-modulating drugs and of efficient tools for the study of chemical biology and drug discovery. The architectures of complex natural products are generally considered to represent significant barriers to efficient chemical synthesis. Here we describe a concise and efficient asymmetric synthesis of 19-dehydroxyl arisandilactone A—which belongs to a family of architecturally unique, highly oxygenated nortriterpenoids isolated from the medicinal plant *Schisandra arisanensis*. This synthesis takes place by means of a homo-Michael reaction, a tandem retro-Michael/Michael reaction, and Cu-catalysed intramolecular cyclopropanation as key steps. The proposed mechanisms for the homo-Michael and tandem retro-Michael/Michael reactions are supported by density functional theory (DFT) calculation. The developed chemistry may find application for the synthesis of its other family members of *Schisandraceae* nortriterpenoids.

[1] Key Laboratory of Bioorganic Chemistry and Molecular Engineering of Ministry of Education, Beijing National Laboratory for Molecular Science (BNLMS), College of Chemistry and Molecular Engineering, Peking-Tsinghua Center for Life Sciences, Peking University, Beijing 100871, China. [2] School of Chemistry and Chemical Engineering, Chongqing University, Chongqing 400030, China. [3] Laboratory of Chemical Genomics, School of Chemical Biology and Biotechnology, Peking University Shenzhen Graduate School, Shenzhen 518055, China. [4] Key Laboratory of Marine Drugs, Chinese Ministry of Education, School of Medicine and Pharmacy, Ocean University of China, 5 Yushan Road, Qingdao 266003, China. * These authors contributed equally to this work. Correspondence and requests for materials should be addressed to Y.L. (email: lanyu@cqu.edu.cn) or to J.-H.C. (email: jhchen@pku.edu.cn) or to Z.Y. (email: zyang@pku.edu.cn).

*S*chisandra chinensis is also known as wu wei zi in China, which translates to five-flavour berry, a description given to it because it possesses all five basic flavours of salty, sweet, sour, spicy and bitter. Records show that for over 2,000 years five-flavour berries have been used as sedatives and tonic agents, as well as for the treatment of rheumatic lumbago and related diseases[1,2].

Because *S. chinensis* is a traditional Chinese herbal medicine, it has been targeted in medicinal chemistry to identify lead compounds for drug discovery. There has been considerable progress in the discovery of bioactive triterpenoids from the *Schisandraceae* family over the past two decades. To date, over 200 nortriterpenoids have been structurally characterized[1,2], and some representative structures **1–6** are shown in Fig. 1. Preliminary biological assays have indicated that some of the nortriterpenoids possess inhibitory activity toward hepatitis, tumours and HIV-1 (refs 1,2).

Natural sources of these compounds are scarce, which hampers systematic studies of their biological activities. Synthesis of these products is needed to continue biomedical research in this area, and much effort has been devoted to total synthesis of these nortriterpenoids ('Synthetic studies on Schisandraceae triterpenoids', see ref. 3 and related references therein). From this research, total syntheses of schindilactone A[4], rubriflordilactone A[5,6], schilancitrilactones B and C[7] and propindilactone G[8] (see **3–6** in Fig. 1) have been developed.

In 2010, Shen isolated (+)-arisandilactone A[9] and (+)-arisanlactone C[10] (**1** and **2** in Fig. 1) from *S. arisanensis*, which is found in Taiwan. The structures of these compounds were confirmed by X-ray crystallography. Among the nortriterpenoids, arisandilactone A (**1**) is unique in that it has an oxa-bridged 7-9-5 tricyclic carbon core, which has not been encountered in natural products before. Because of this structural feature, arisandilactone A (**1**) is a challenging target for total synthesis. To the best of our

knowledge, no synthesis of arisandilactone A (**1**) has been reported to date.

Natural products with unique and complex architectures are challenging in organic synthesis (for an excellent review, see ref. 11). In this regard, biomimetic approaches for the formation of complex natural products have proven to be effective strategies in organic synthesis[12]. Among the various biomimetic reactions, Michael-type reaction can be regarded as one of the most powerful reactions in the biosynthesis of natural products[13]. Here, we report an asymmetric total synthesis of (+)-19-dehydroxyl arisandilactone A (**1a**), a derivative of arisandilactone A (**1**), using a biomimetic synthesis involving a homo-Michael reaction and a tandem retro-Michael/Michael reaction as key steps.

## Results

**Retrosynthetic analysis.** For the total synthesis of arisandilactone A (**1**), construction of its strained transannular 10-oxabicyclo[5.2.1]dec-6-en-8-one core (DE ring) bearing an oxa-bridged α,β-unsaturated ketone subunit is the most difficult task in the entire total synthesis. Thus, from a synthetic strategy point of view, this fragment should be introduced at a late stage in the total synthesis, and preferable by a biomimetic synthesis because of its high sensitivity and instability.

While conducting the model study toward the total synthesis of arisanlactone C (**2**), we serendipitously found that this strained bicyclic core could be constructed via an intramolecular oxa-homo-Michael reaction (Fig. 2). In the event, we first synthesized aldehyde **7** and then coupled with tert-butyldimethyl((3-methyl-furan-2-yl)oxy)silane (**8**) via a vinylogous Mukaiyama aldol reaction (VMAR). To our surprise, product **9** was obtained in approximately 50% yield when the reaction was carried out at − 78 °C in the presence of BF$_3$·Et$_2$O in CH$_2$Cl$_2$. The reaction

**Figure 1 | *Schisandraceae* family.** Naturally occurring nortriterpenoids (**1–6**) and 19-dehydroxyl arisandilactone A (**1a**, a derivative of **1**).

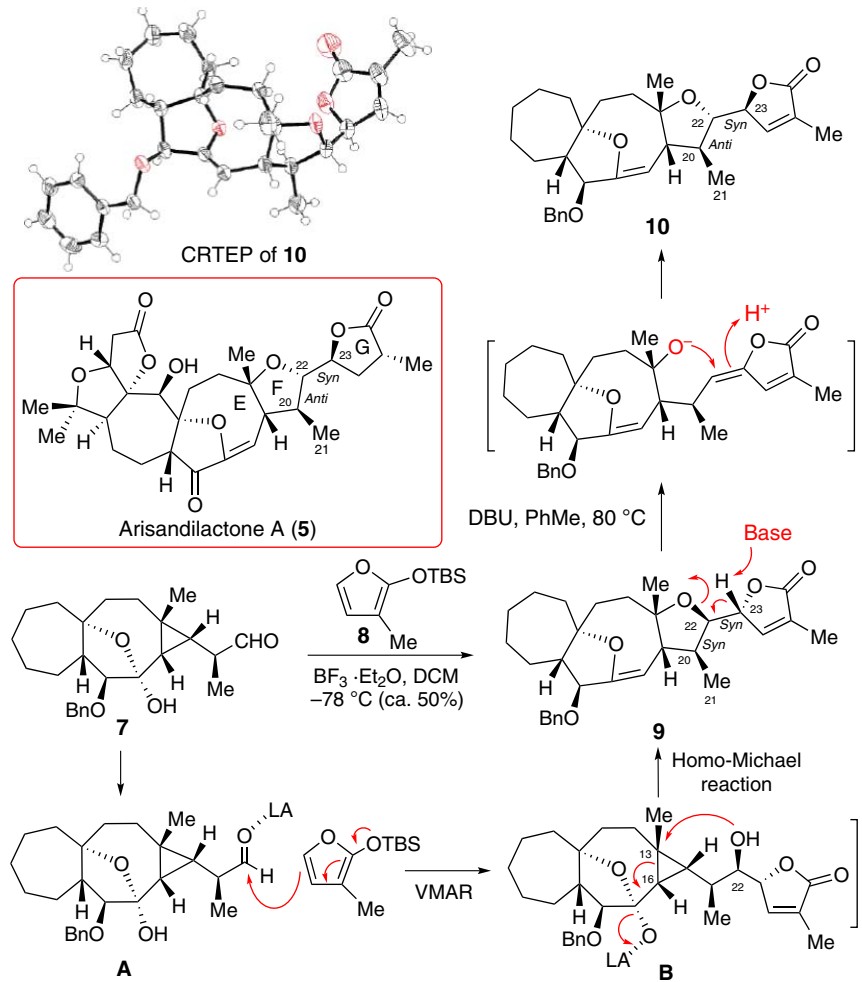

**Figure 2 | Model study.** Unified biosynthetic approach featuring homo-Michael and retro-Michael/Michael reactions for formation of the 7/9/5 tricyclic core of arisandilactone A (**1**).

was proposed to proceed through a typical VMAR to form intermediate **B**. This was followed by a homo-Michael reaction[14,15] through the cleavage of its carbon–carbon bond at C13 and C16 and simultaneous formation of the five-membered F ring (see Fig. 2). The newly generated three stereogenic centres at C20, C22 and C23 in product **9** formed a *syn/syn* stereotriad. The arrangements of groups at the stereogenic centres at C22 and C23 were opposite to those in arisandilactone A, which has an *anti/syn* stereotriad. Stereoselective synthesis of the *anti/syn* stereotriad has proven to be challenging[16,17].

On inspection of the structure of product **9**, we found that **9** could be epimerized at its C22 and C23 stereogenic centres by treatment with 1,8-diazabicyclo[5.4.0]undec-7-ene (DBU) via a tandem retro-Michael/Michael reaction. This strategy has been used by Pyne and co-workers in the total synthesis of *Stemona* alkaloid (11*S*, 12*R*)-dihydrostemofoline (**C**) from (11*S*, 12*S*)-dihydrostemofoline (**A**) through intermediate **B** (Fig. 3)[18]. The stereochemical outcome of the DBU-initiated ring opening reaction of **A** can be rationalized as occurring through a deprotonation of **A** by DBU at the acidic γ-position of the lactone ring which would result in the anionic intermediate **B**, and then lead to (11*S*, 12*R*)-dihydrostemofoline (**C**) to avoid the unfavourable steric interaction between the methoxy (C13) and methyl (C10) groups in the intermediate **B** (Fig. 3).

To explore the feasibility of this strategy, we treated **9** with DBU in toluene at 80 °C. As expected, the desired product **10** was

obtained in 48% yield (Fig. 2), and its structure was tentatively confirmed through X-ray crystallography; however, the data were of insufficient quality to allow a definitive determination of the structure. Inspired by these results, we applied this knowledge to the total synthesis of 19-dehydroxyl arisandilactone A (**1a**) using the homo-Michael and the tandem retro-Michael/Michael reactions as key steps.

Figure 4 illustrates our proposed biomimetic total synthesis of 19-dehydroxyl arisandilactone A. Based on the chemistry presented above, we expected the target **1a** could be produced from precursor **A** using the developed tandem retro-Michael/Michael reaction as a key step. The intermediate **A**, in turn, could be produced from **B** via nucleophilic attack of the silyl ether **C** on the aldehyde in intermediate **B** via a VMAR, followed by the key intramolecular homo-Michael reaction as shown above. Precursor aldehyde **B** could be prepared from lactone **D** via a series of functional group interconversions.

It is conceivable that the *cis*-fused 8/3-bicyclic domain in lactone **D** could be constructed via a Cu-catalysed intramolecular cyclopropanation[19] of **E** by taking advantage of conformation control of the substrate. This reaction would generate **D** with three contiguous stereogenic centres at C13, C16 and C17, as well as a lactone fragment, which could serve as a handle to install the side chain. To the best of our knowledge, synthesis of a *cis*-fused 8/3-bicyclic ring fragment in **D** is unprecedented. Thus, our retrosynthetic analysis could trace back to the synthesis of the

**Figure 3 | retro-Michael/Michael reaction.** DBU mediated a biomimetic tandem retro-Michael/Michael reaction for the formation of (11S, 12R)-dihydrostemofoline (**C**) from (11S, 12S)-dihydrostemofoline (**A**) through intermediate **B**.

**Figure 4 | Retrosynthetic analysis of 19-dehydroxyl arisandilactone A (1a).** The Homo-Michael/Michael reaction were used as key steps to construct core structure of 19-dehydroxyl arisandilactone A (**1a**).

oxa-bridged 'tri-substituted' cyclooctene ring in **E**, which could be generated from diene **F** using a ring-closing metathesis (RCM) reaction, a method that was utilized in our total synthesis of schindilactone A[5] (**3** in Fig. 1). Diene **F**, in turn, could be made from ketoester **G** by modifying the methods applied in our total synthesis of schindilactone A. Thus, (R)-( − )-carvone (**H**) is a logical starting material for the preparation of ketoester **G**.

**Asymmetric synthesis of 19-dehydroxyl arisandilactone A (1a).** Our synthesis began with asymmetric construction of the fully functionalized bislactone **16** from (R)-( − )-carvone (Fig. 5).

A two-step procedure for the conversion of (R)-( − )-carvone to cycloheptenone was accomplished using a Mander's methoxycarbonylation, followed by a sequence of cyclopropanation/ring-expansion reactions[20,21] to afford **11** in 68% yield. Compound **11** was hydrolysed with LiOH, and the resulting acid then underwent an AgOTf-mediated lactonisation[22] to afford **12** in 66% overall yield. To install the C10 silyloxy group of **13**, **12** was first exposed to TMSO(CH$_2$)$_2$OTMS/TMSOTf[23], and the resultant ketal ester was then reacted with potassium bis(trimethylsilyl)amide in the presence of P(OMe)$_3$ under an O$_2$ atmosphere[24], followed by reaction with triethylchlorosilane to afford **13** in 70% yield as a single diastereoisomer.

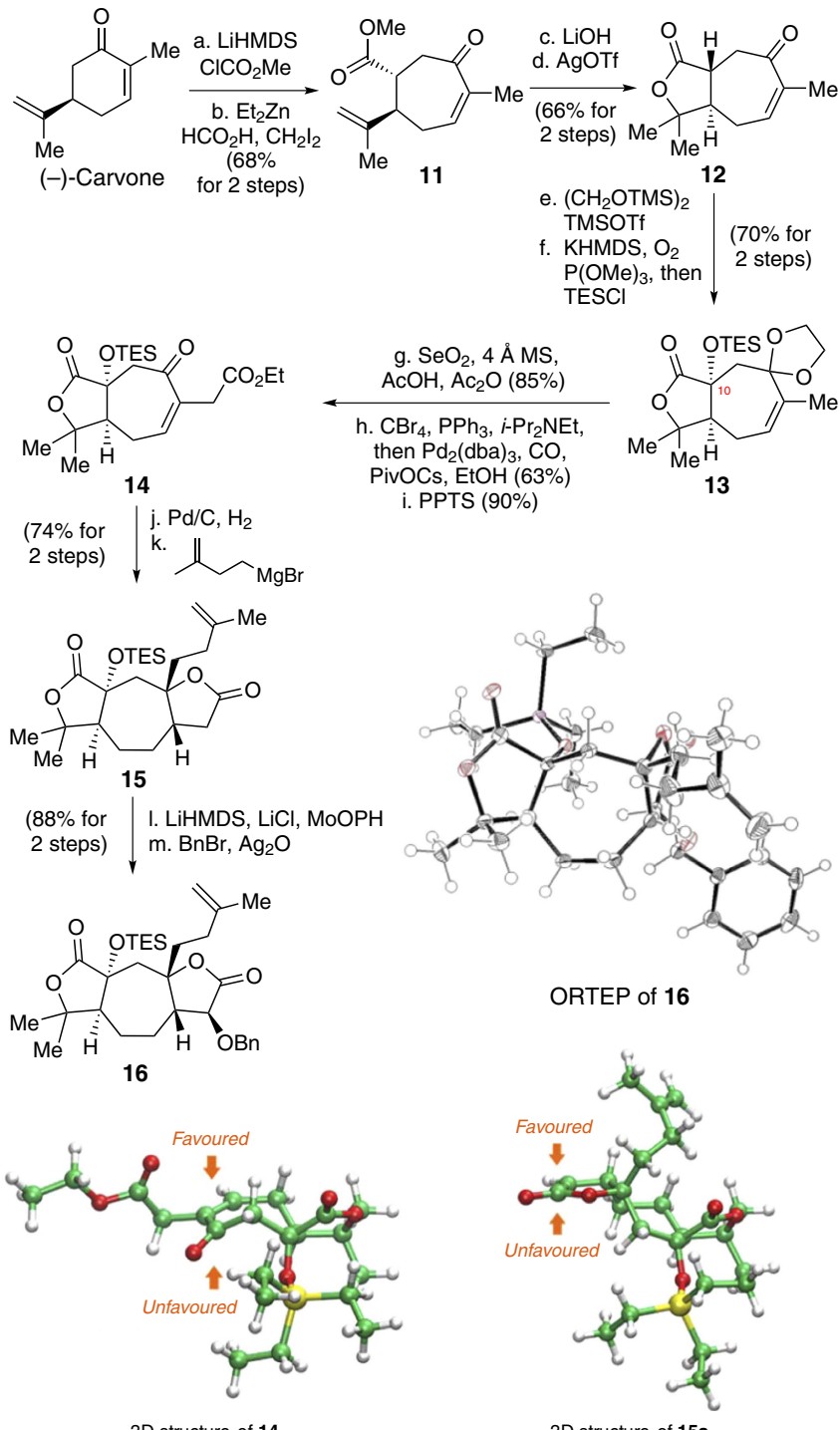

**Figure 5 | Synthesis of compound 16.** (a) ( − )-carvone (1.0 eq), LiHMDS (2.1 eq), ClCOOMe (1.5 eq), THF, − 78 °C, 2 h, 90%; (b) Et$_2$Zn (2.3 eq),
CH$_2$I$_2$ (2.3 eq), HCOOH (2.3 eq), DCM, 0 °C, 75%; (c) LiOH (2.2 eq), MeCN/H$_2$O (1:1), rt, 92%; (d) AgOTf (0.05 eq), ClCH$_2$CH$_2$Cl, reflux, 72%;
(e) TMSOCH$_2$CH$_2$OTMS (1.4 eq), TMSOTf (0.1 eq), DCM, − 78 to − 20 °C, 87%; (f) KHMDS (2.0 eq), P(OMe)$_3$ (1.6 eq), − 78 °C, THF, under a balloon
filled with O$_2$, then TESCl (1.4 eq), 81%; (g) SeO$_2$ (3.0 eq), Ac$_2$O (1.0 eq), AcOH (0.16 eq), 4 Å MS (120% mass fraction), dioxane, 65 °C, 85%; (h) CBr$_4$
(1.2 eq), PPh$_3$ (1.2 eq), $^i$Pr$_2$NEt (0.4 eq), DCM, 0 °C, 3 h, then Pd$_2$(dba)$_3$ (0.05 eq), PivOCs (0.5 eq), $^i$Pr$_2$NEt (0.8 eq), EtOH, under a balloon filled with CO,
0 °C to rt, 63%; (i) PPTS (0.1 eq), acetone/H$_2$O (10:1), 60 °C, 90%; (j) Pd/C (10% mass fraction), EtOAc, under balloon filled with H$_2$, rt, 92%; (k) Grignard
reagent (3.0 eq), THF, − 20 °C, 80%; (l) LiHMDS (2.0 eq), LiCl (2.0 eq), MoOPH (1.5 eq), THF, − 78 °C, 91%; and (m) BnBr (2.0 eq), Ag$_2$O (2.0 eq), DCM, rt,
97%. LiHMDS, lithium hexamethyldisilylamide. KHMDS, potassium hexamethyldisilylamide. MoOPH, oxodiperoxymolybdenum(pyridine)(hexamethylphos-
phoric triamide), also refered to as Vedejs' reagent; 4 Å MS, molecular sieves, type 4A; Pd$_2$(dba)$_3$, tris(dibenzylideneacetone)dipalladium(0); PivOCs, cesium
pivolate; PPTS, pyridium p-toluenesulfonate; TESCl, triethylsilyl chloride.

**Figure 6 | Synthesis of the lactone 20.** (a) Grignard reagent (1.5 eq), THF, rt; (b) Hoveyda-Grubbs II (**17**, 0.05 eq), PhMe, 80 °C, 79% in two steps; (c) Ac$_2$O (2.0 eq), DMAP (0.5 eq), Et$_3$N (5.0 eq), DCM, rt, 97%; (d) LiHMDS (2.5 eq), CF$_3$CO$_2$CH$_2$CF$_3$ (1.5 eq), THF, −78 °C to 0 °C. (e) TsN$_3$ (1.4 eq), Et$_3$N (3.0 eq), MeCN, rt, 93% in two steps; (f) Cu(tbs)$_2$ (0.1 eq), PhMe, 80 °C, 4.5 h, 65%. ; Cu(tbs)$_2$, bis(N-tert-butylsalicylaldiminato) copper(II); DMAP, 4-dimethylaminopyridine; TsN$_3$, tosylazide.

Next, we treated **13** with SeO$_2$ in the presence of AcOH[25], Ac$_2$O and 4 Å molecular sieves for an allylic oxidation. The resulting allylic alcohol was then converted to its corresponding carboxylic ester by bromination[26] and Pd-catalysed carbonylation[27] to give an ester in 63% yield in one step. After deprotection, enone **14** was obtained in 90% yield. To synthesize bislactone **16** from enone **14**, we adopted the strategy developed in our total synthesis of schindilactone A[5]. Enone **14** was sequentially subjected to a Pd-catalysed hydrogenation (92%) and Grignard reaction (80%) to give **15** in 74% yield as a single diastereoisomer. The observed diastereoselectivity could be attributed to the steric bulk of the triethylsilyloxy group in substrate **14**, which allows both the hydrogenation and Grignard reactions to occur at the less-hindered face (see three-dimensional structure of **14** in Fig. 5)

To achieve facially selective α-hydroxylation, bislactone **15** was treated with LiHMDS in the presence of LiCl[28] at −78 °C in THF. The resulting enolate complex was subsequently oxidized with MoOPH[29] to afford a secondary alcohol, which was then treated with BnBr/Ag$_2$O[30] to give bislactone **16** in 88% overall yield as a single diastereoisomer. The relative stereochemistry of **16** was confirmed by X-ray crystallography (Fig. 5).

This highly stereoselective α-hydroxylation was also likely promoted by the steric effect of the triethylsilyloxy group in enolate **15a**, which was generated in situ during the treatment of **15** with LiHMDS, forcing the oxidant to approach from the less-hindered face (see three-dimensional structure of **15a** in Fig. 5).

We then investigated our proposed RCM reaction to construct the medium-sized ring-based[31] tri-substituted cyclooctene in compound **18** (Fig. 6). By screening various reaction conditions (for example, catalysts, solvents, additives and reaction temperatures), we determined that the use of the second-generation Hoveyda–Grubbs catalyst (**17**) in toluene at 80 °C afforded the best results. Considering the potential for in situ epimerization of hemiketal during the RCM reaction[5,32], we treated bislactone **16** with vinyl magnesium bromide[33]. The resulting dienes, as an epimeric mixture of hemiketals, were then subjected to a RCM reaction in the presence of catalyst **17** (ref. 34; 5 % amount-of-substance fraction) in toluene at 80 °C for 18 h. To our delight,

the desired product **18** was obtained as a single diastereoisomer in an overall 79% yield for the two steps.

For stereoselective formation of the 8/3 bicyclic-ring system in **20**, an efficient method was required to prepare α-diazoacetate **19** from hemiketal **18**. Initial attempts to prepare **19** by esterification coupling of **18** with ClCOCH = NNHTs[35] in the presence of a base produced only a trace amount of the desired product. This could be attributed to steric hindrance at the C15 hydroxyl group in **18**. Consequently, we carried out a systematic study for the synthesis of **19**. We found out that **19** could be effectively generated by the treatment of **18** with Ac$_2$O/Et$_3$N in the presence of DMAP followed by reaction with LiHMDS. The resulting enolate was reacted with CF$_3$CO$_2$CH$_2$CF$_3$ at −78 °C, and then treated with TsN$_3$ to afford **19** in an overall 90% yield for the three steps. We next investigated the conditions for synthesis of the rigid and highly strained intermediate **20** from **19** via the proposed intramolecular cyclopropanation reaction. To date, intramolecular cyclopropanation for the formation of an 8/3 fused bicyclic ring system has not been reported. Consequently, it was unknown whether such a rigid and highly strained intermediate could be generated. We hypothesized that strategical positioning of α-diazoacetate could be used to facilitate the cyclopropanation in a diastereoselective manner. To achieve this goal, we systematically profiled the cyclopropanation reaction with different metal catalysts, including CuSO$_4$, CuCl, bis(2,4-pentanedionato) copper(II) (Cu(acac)$_2$), CuI, Rh$_2$(OAc)$_4$ and Cu(tbs)$_2$. We found that the Cu(tbs)$_2$-catalysed cyclopropanation[36] of **19** afforded **20** in 65% yield as a single diastereoisomer. The structure of **20** was confirmed by two-dimensional NMR spectroscopy.

With lactone **20** in hand, we next investigated its transformation to aldehyde **27**. As illustrated in Fig. 7, the first synthetic task was the conversion of the lactone ring in **20** to the terminal olefin in **21**. Exposure of **20** to a solution of N,O-dimethylhydroxylamine hydrochloride and isopropylmagnesium chloride in THF at −20 °C (ref. 37) formed a Weinreb amide in 90% yield, which was reacted with methylmagnesium chloride to afford a ketone in 89% yield. This ketone then underwent Peterson olefination[38] by reaction with freshly produced (trimethylsilyl)methylmagnesium

**Figure 7 | Synthesis of alcohol 27.** (a) MeNHOMe · HCl (5.0 eq),$^i$PrMgCl (10.0 eq), THF, −20 °C, 90%; (b) MeMgCl (3.0 eq), THF, 0 °C, 89%; (c) TMSCH$_2$MgCl (5.0 eq), CeCl$_3$ (5.0 eq), THF, 0 °C, then silica gel, DCM, 83%; (d) TBAF (3.0 eq), THF, 0 °C, 99%; (e) TMS-imid. (10.0 eq), DCM, 0 °C, 95%; (f) Ac$_2$O (15.0 eq), DMAP (1.0 eq), Et$_3$N (15.0 eq), PhMe, rt, 98%; (g) LiHMDS (3.0 eq), THF, −78 to −35 °C, 93% b.r.s.m.; (h) Martin's sulfurane (1.2 eq), DCM, rt; (i) L-selectride (3.0 eq), −78 °C; (j) TBAF (5.0 eq), THF, 0 °C, 93% in three steps; (k) BMS (10.0 eq), THF, 0 °C, 0.75 h, then Na$_2$B$_4$O$_7$ (excess), H$_2$O$_2$ (excess), rt, 65%; (l) TPAP (1.3 eq.), DCM, 0 °C, 88%. BMS, borane-dimethylsulfide complex; b.r.s.m., based on recovered starting material; L-selectride, lithium tri-sec-butylborohydride; Martin's sulfurane, bis[α,α-bis(trifluoromethyl) benzenemethanolato]- diphenylsulfur; TBAF, tetra-n-butylammonium fluoride; TMS-imid., 1-(trimethylsilyl)-1H-imidazole; TPAP, tetra-n-propylammonium perruthenate(VII).

chloride in the presence of CeCl$_3$ (ref. 39) in THF at 0 °C, and the resultant mixture was worked up by the treatment with silica gel to give an olefin, which, without purification, was then subjected to a desilylation with TBAF to give **21** in 82% yield for the two steps. The structure of **21** was confirmed by X-ray crystallography.

It is worthwhile to mention that when the ketone derived from the Weinreb amide was directly reacted with a Wittig reagent derived from methyltriphenylphosphonium bromide, the expected olefination proceeded. However, epimerization occurred at C17 of the product **21**.

The next task in the synthesis of aldehyde **27** was A ring formation in intermediate **24**. Diol **21** was selectively protected by sequential treatment with TMS-imid. and Ac$_2$O/DMAP, to give **22** in an overall 93% yield for the two steps. Dieckmann-type condensation[4] of **22** proceeded smoothly to afford product **23**

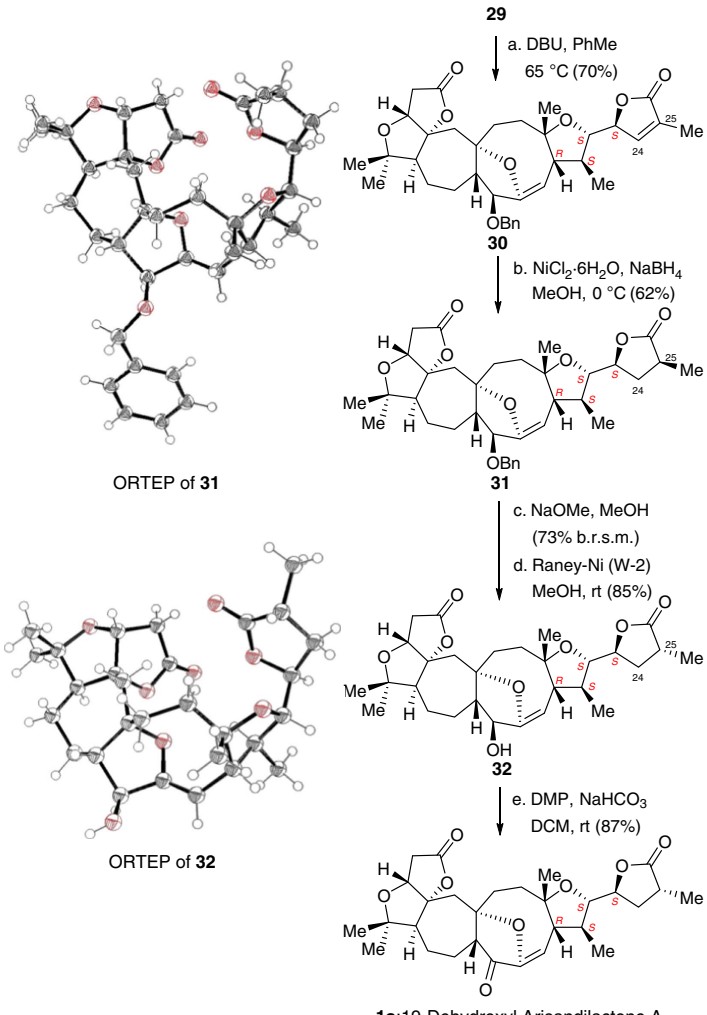

**Figure 8 | Total synthesis of 29.** (a) **28** (3.0 eq), BF$_3$·OEt$_2$ (3.3 eq), DCM, −78 to −35 °C, 66%.

**Figure 9 | Total synthesis of 19-dehydroxyl-arisandilactone A (1a).** (a) DBU (15.0 eq), PhMe, 65 °C, 70%; (b) NiCl$_2$·6H$_2$O (1.0 eq), NaBH$_4$ (3.0 eq), MeOH, 0 °C, 62%; (c) 5% NaOMe/MeOH, rt, 73% (b.r.s.m.); (d) W-2 Raney Ni, EtOH, rt, 85%; (e) DMP (2.0 eq), NaHCO$_3$ (5.0 eq), DCM, rt, 87%. b.r.s.m., based on recovered starting material; DMP, Dess-Martin periodinane.

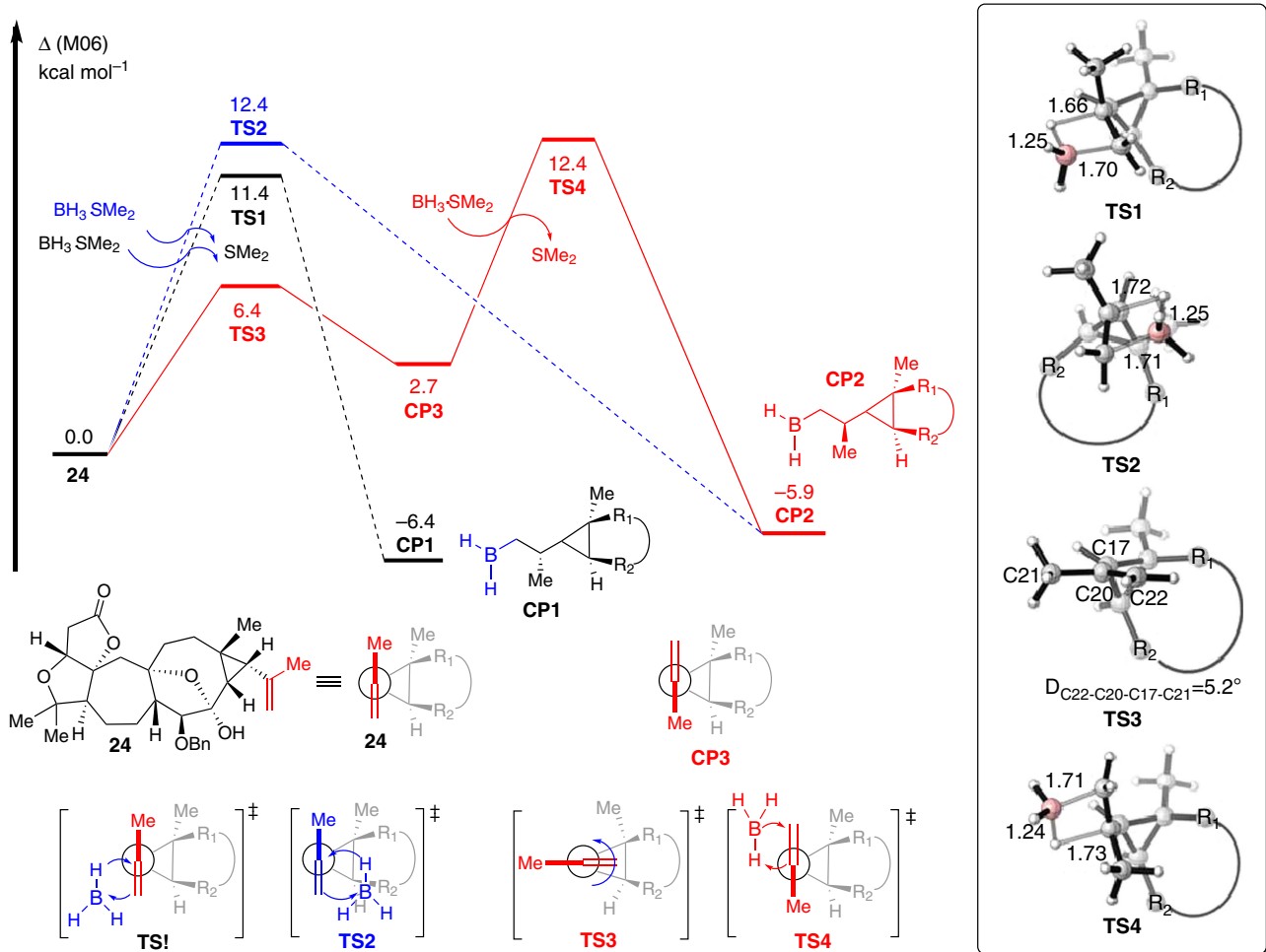

**Figure 10 | Energy profiles for the hydroboration of 24.** The values given by kcal mol$^{-1}$ are the relative free energies calculated by M06 method in tetrahydrofuran solvent.

in 93% yield. Dehydration of the C1 hydroxyl group in **23** was achieved by treatment with Martin's sulfurane[40], and the resultant α,β-unsaturated lactone was reduced with L-selectride[41], followed by desilylation to give **24** as a single diastereoisomer in 93% yield for three steps.

The synthesis of alcohol **26** proved challenging because **24** contained a lactone and a highly rigid three-membered ring-linked olefin[42]. These groups are sensitive to both hydroboration and basic conditions. Thus, after intensive experimentation, we found that treatment of **24** with BMS[43] in THF at 0 °C, followed by oxidation with a solution of H$_2$O$_2$ (30%) in the presence of a weak base (Na$_2$B$_4$O$_7$; ref. 44), product **26** could be obtained in 65% yield, together with its diastereoisomer (16% yield). The stereochemistry at C20 in **26** was confirmed by X-ray crystallography. Thus, synthesis of aldehyde **27** could be achieved in 88% yield by oxidation of substrate **26** with TPAP.

With synthesis of aldehyde **27** achieved, we attempted the proposed biomimetic synthesis of 19-dehydroxyl arisandilactone A (**1a**) (Fig. 8). In the event, **27** was coupled with **28** in the presence of BF$_3$·Et$_2$O in CH$_2$Cl$_2$ at −78 °C, and the resultant mixture was then stirred at −35 °C for 1 h, followed by workup to give product **29** in 66% yield as a single diastereoisomer. The reaction actually proceeded through a typical VMAR followed by the proposed homo-Michael reaction. The structure of **29** was confirmed by X-ray crystallography, which indicated that the stereogenic centers at C22 and C23 were opposite to those in the natural product.

We now move to the stage for the completion of the total synthesis. To this end, substrate **29** was treated with DBU in toluene at 65 °C, as a result, product **30** was obtained in 70% yield with inversion of the stereogenic centres at C22 and C23 (Fig. 9). Thus, further treatment of **30** with NiCl$_2$·6H$_2$O/NaBH$_4$, the double bonds at C24 and C25 in **30** could be chemo-selectively saturated to give product **31** in 62% yield. However, X-ray crystallographic analysis of **31** indicated that its stereochemistry at C25 was opposite to that in the natural form. We, therefore, treated **31** with NaOMe in methanol to invert its stereochemistry at C25, and the resultant product was then subjected to Raney nickel-mediated debenzylation to give product **32** in 73% yield and 85% yield, respectively. The structure of **32** was confirmed by X-ray crystallography. Thus, our total synthesis of 19-dehydroxyl arisandilactone A (**1a**) was eventually achieved in 87% yield by oxidation of **32** with DMP. Overall, this asymmetric synthesis consists of 37 steps in its longest linear sequence from (R)-(−)-carvone.

**Computational study**. DFT calculations were carried out using the Gaussian 09 program (Gaussian Inc., Wallingford, CT, USA) to determine the diastereoselectivity for the hydroboration step to stereoselectively form **26** from **24** (see Fig. 10). The method B3LYP with 6-31G(d) basis set was used for geometry optimizations in gas phase. M06 functional with larger basis set 6-311 + G(d) was used for solvation single point calculations

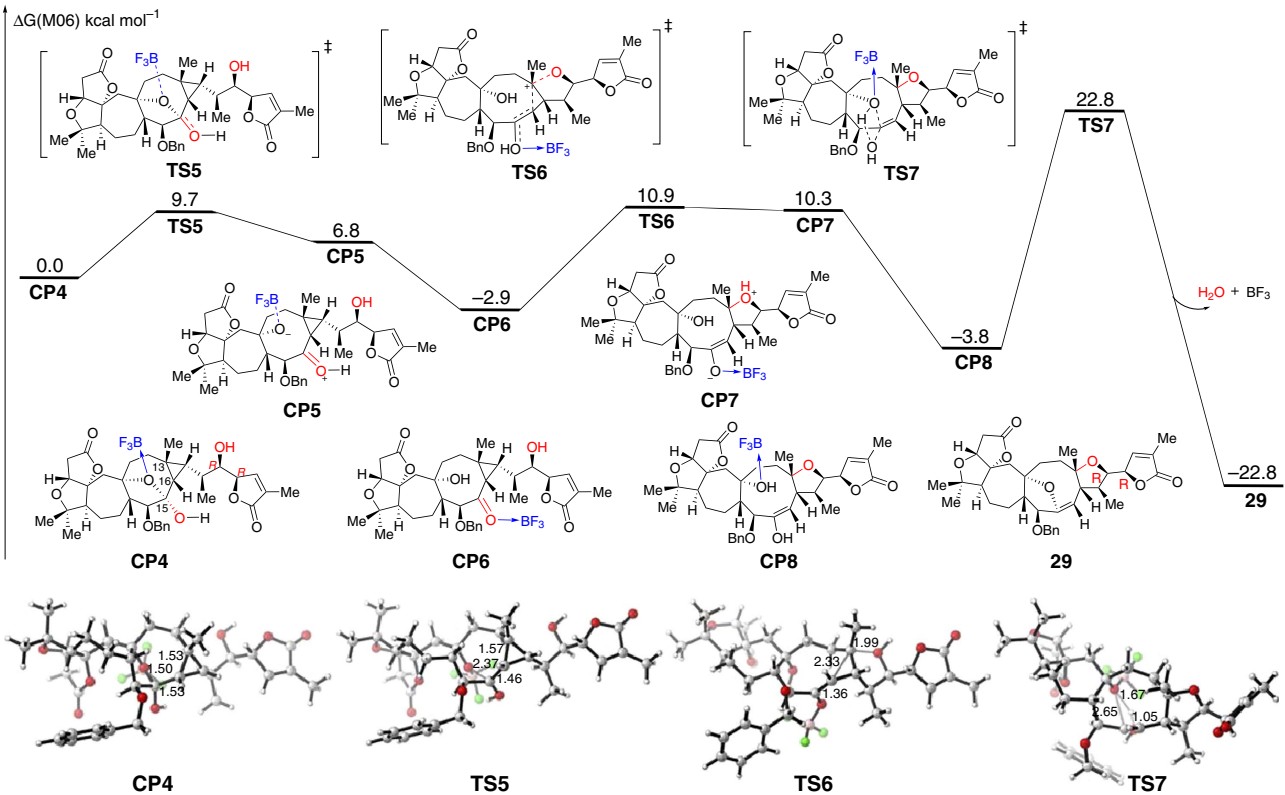

**Figure 11 | Energy profiles for the homo-Michael step.** The values given by kcal mol$^{-1}$ are the relative free energies calculated by M06 method in dichloromethane solvent.

in THF based on gas-phase stationary points using the continuum solvation model SMD (see Supplementary Methods for details).

As shown in Fig. 10, olefin **24** was set to relative zero for the free energy profiles. Initially, the Cossy model[43] was considered to explain the diastereoselectivity. We found that the relative free energy of **TS1** was 1.0 kcal mol$^{-1}$ lower than that of **TS2**, which could be attributed to the repulsion between the borane moiety and the substituents on cyclopropane in the transition state **TS2**. The diastereoisomer **CP1**, which is the precursor of alcohol **26**, is the major product generated via transition state **TS1**. This result is consistent with experimental observations. Moreover, we also found that rotation of an alkenyl could take place via transition state **TS3** with a barrier of only 6.4 kcal mol$^{-1}$. The relative free energy of the rotational isomer **CP3** was 2.7 kcal mol$^{-1}$ higher than that of compound **24**. This was attributed to the larger size of R$_2$ compared with R$_1$, and the closer positioning of R$_2$ to the methyl group in **CP3**. However, when hydroboration takes place from **CP3** via transition state **TS4**, the hybridization of C22 changes from $sp^2$ to $sp^3$, and the size difference between the methyl and methylene groups is reduced. Therefore, the relative free energy of **TS4** was only 1.0 kcal mol$^{-1}$ higher than that of **TS1**. The DFT calculations indicated that favourable pathway for the generation of major intermediate **CP1** takes place via the same transition state **TS1** in both the Cossy model and our proposed reaction model. However, there are two different pathways for the generation of side intermediate **CP2** including the Cossy model via transition state **TS2** and our proposed model via transition state **TS4**, the barriers for which are closed.

We applied the same method to study the mechanism of the homo-Michael reaction. As shown in Fig. 11, complex **CP4**, which is formed by reaction between aldehyde **27** and silyl ether

**28** with the coordination of BF$_3$ is set to the relative zero for the free-energy profiles of this reaction. With the activation of BF$_3$, the C–O bond cleavage forms a protonated ketone intermediate **CP5** via transition state **TS5** with a barrier of only 9.7 kcal mol$^{-1}$. The subsequent isomerization generates intermediate **CP6** with 9.7 kcal mol$^{-1}$ exothermic. With the activation of BF$_3$ · Et$_2$O, C13 and C16 is more electron-deficient. Therefore, the nucleophilic addition with hydroxyl takes place via transition state **TS6** with a barrier of 13.8 kcal mol$^{-1}$. After a proton transfer, an enol intermediate **CP8** is formed. The followed cyclization occurs via transition state **TS7** with a barrier of 26.6 kcal mol$^{-1}$. After releasing of one molecule water coordinated BF$_3$, product **29** is formed exothermically.

Finally, we used the same method to study the mechanism of the isomerization of **29** to **30**. As shown in Fig. 12, the intermolecular electrophilic deprotonation by DBU of substrate **29**, which gives intermediate **CP9** endothermically, occurs via transition state **TS8** with a barrier of 28.2 kcal mol$^{-1}$. Subsequently, the retro-Michael addition takes place via transition state **TS9** to break the C–O bond, with the help of DBU-H$^+$ to stabilize the formed ionic oxygen atom, to generate intermediate **CP10** reversibly with a barrier of 13.3 kcal mol$^{-1}$.

In **CP10** rotation of the C22 − C20 single bond affords **CP11** in a 2.3 kcal mol$^{-1}$ exothermic step. The Michael addition takes place at the Si-face of the alkene moiety in **CP11** via transition state **TS10**, with a barrier of 11.4 kcal mol$^{-1}$, and generates intermediate **CP12** reversibly. The direct protonation of **CP12** by DBU-H$^+$ could occur via transition state **TS13** or **TS14**.

However, **30** was not the major product because the relative free energy of **TS14**, which leads to the formation of **CP14** was 1.0 kcal mol$^{-1}$ lower than that of **TS13** which leads to **30**. Alternatively, we found that a single C–C bond in **CP12** could rotate rapidly via transition state **TS11**, and isomer **CP13** could

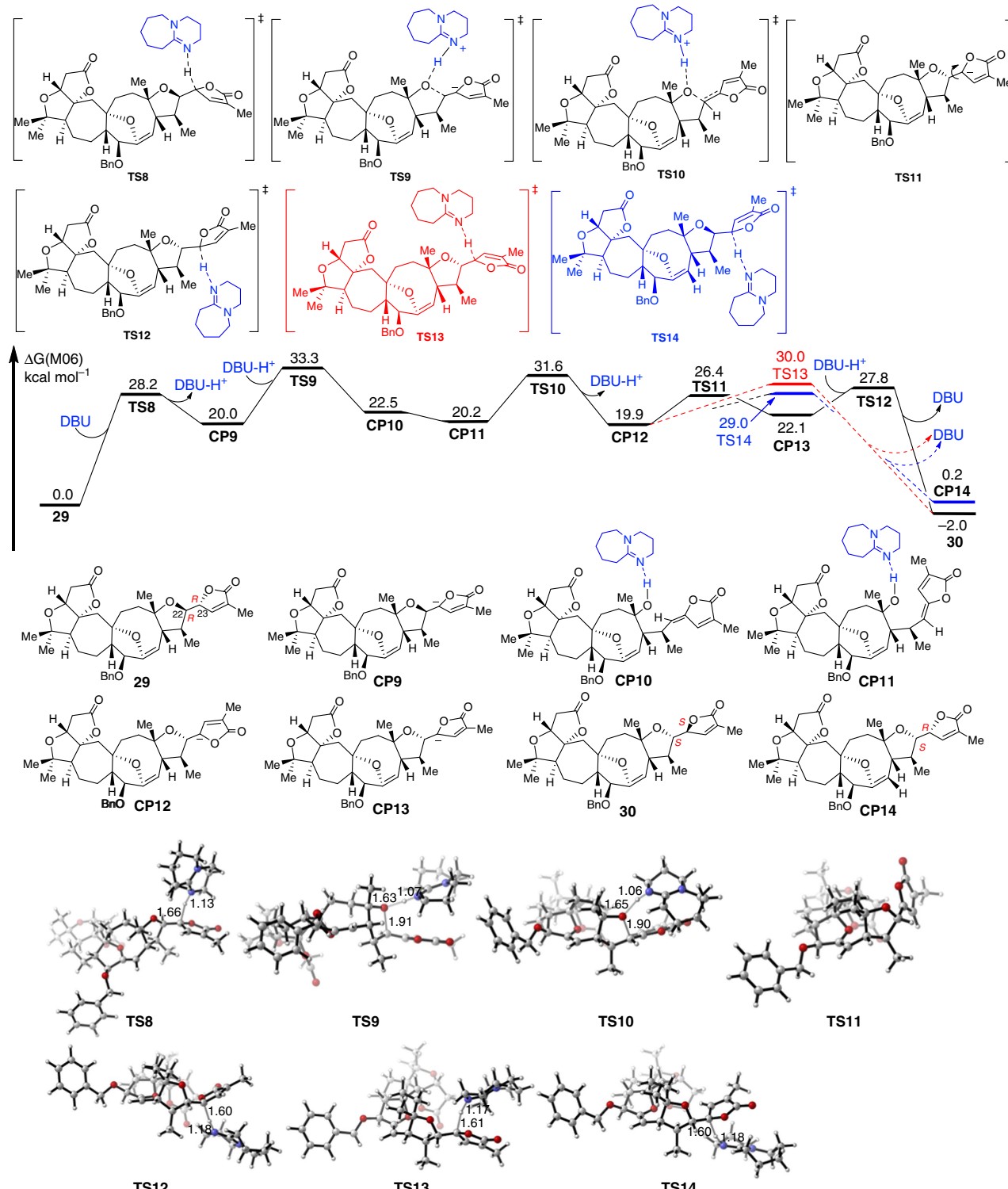

**Figure 12 | Energy profiles for the reversible retro-oxa-Michael reaction of 29.** The values given in kcal mol$^{-1}$ are the relative free energies calculated by the M06 method in tetrahydrofuran solvent.

be formed reversibly. The protonation of **CP13** via transition state **TS12** could also afford product **30**, and the relative free energy of **TS12** was 1.2 kcal mol$^{-1}$ lower than that of **TS14**, Moreover, the relative free energy of **30** was 2.2 kcal mol$^{-1}$ lower than that of **CP14**, and 2.0 kcal mol$^{-1}$ lower than that of **29**. Therefore, **30** was also a thermodynamically stable product, which was observed experimentally.

## Discussion

In conclusion, we successfully developed an asymmetric total synthesis of (+)-19-dehydroxyl arisandilactone A (**1a**). Construction of an oxa-bridged and moderately sized DE ring involved an unprecedented tandem VMAR and oxa-homo-Michael type reaction. Stereoselective formation of stereogenic centres at C22 and C23 of **1a** were achieved by a tandem and

biomimetic retro-Michael/Michael reaction. Key steps in this total synthesis also included a RCM reaction for the diastereoselective formation of the fully functionalized eight-membered DE ring system, and an interesting Cu-catalysed intramolecular cyclopropanation that resulted in stereoselective formation of the central 8/3 *cis*-fused bicyclic ring system. The structure of 19-dehydroxyl (+)-arisandilactone A (**1a**) was confirmed by X-ray crystallography of its precursor alcohol **32**. Details of the reactions for the hydroboration, homo-Michael, and tandem retro-Michael/Michael reactions were determined by DFT calculations. The biological investigation of the synthesized (+)-19-dehydroxyl arisandilactone A (**1a**), as well as the advanced intermediates in this total synthesis is currently underway in our laboratory.

## Methods

**General**. All reactions were carried out under a nitrogen atmosphere under anhydrous conditions and all reagents were purchased from commercial suppliers without further purification. Solvent purification was conducted according to *Purification of Laboratory Chemicals* (Peerrin, D. D.; Armarego, W. L. and Perrins, D. R., Pergamon Press: Oxford, 1980). Yields refer to chromatographically and spectroscopically ($^1$H NMR) homogeneous materials. Reactions were monitored by Thin Layer Chromatography on plates (GF254) supplied by Yantai Chemicals (China) visualized by ultraviolet or stained with ethanolic solution of phosphomolybdic acid and cerium sulfate, basic solution of KMnO₄ and iodine vapour. If not specially mentioned, flash column chromatography was performed using E. Merck silica gel (60, particle size 0.040–0.063 mm). NMR spectra were recorded on Bruker AV400, Bruker AV500 instruments and calibrated by using residual undeuterated chloroform ($\delta$H = 7.26 p.p.m.) and CDCl₃ ($\delta$C = 77.16 p.p.m.), partially deuterated methylene chloride ($\delta$H = 5.32 p.p.m.) and methylene chloride-d2 ($\delta$C = 53.84 p.p.m.), partially-deuterated methanol ($\delta$H = 3.31 ppm) and methanol-d4 ($\delta$C = 49.00 p.p.m.) as internal references. The following abbreviations were used to explain the multiplicities: s = singlet, d = doublet, t = triplet, q = quartet, br = broad, td = triple doublet, dt = double triplet, dq = double quartet, m = multiplet. Infrared (IR) spectra were recorded on a Thermo Nicolet Avatar 330 FT-IR spectrometer. High-resolution mass spectra were recorded on a Bruker Apex IV FTMS mass spectrometer using ESI (electrospray ionization) as ionization method.

For detailed experimental procedures, see Supplementary Methods. For NMR spectra of the synthesized compounds in this article, see Supplementary Figs 1–34. For the comparison of NMR spectra of the natural arisandilactone A and synthetic 19-dehydroxyl arisandilactone A, see Supplementary Table 1. For ORTEP diagrams for compounds **10**, **16**, **21**, **26**, **29**, **31** and **32**, see Supplementary Figs 35–41. For computational calculation details, see Supplementary Methods and Supplementary Data 1. For calculation results using other methods, see Supplementary Figs 42–44.

**Data availability**. The X-ray crystallographic coordinates for structures reported in this article have been deposited at the Cambridge Crystallographic Data Centre (CCDC), under deposition numbers CCDC 1509144 for compound **10**, CCDC 1508989 for compound **16**, CCDC 1509314 for compound **21**, CCDC 1509138 for compound **26**, CCDC 1507976 for compound **29**, CCDC 1507977 for compound **31**, and CCDC 1507978 for compound **32**. These data can be obtained free of charge from the Cambridge Crystallographic Data Centre via http://www.ccdc.cam.ac.uk/data_request/cif.

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

## Acknowledgements

This work is supported by the National Science Foundation of China (Natural Science Foundation of China (Grant No. 21272015, 21372016, 21472006, 21572009 and 21372266). J.-H. C. and Z. Y. are very grateful to Prof Jian Hao from Beijing University of Chemical Technology, and Prof Fuling Yin from Peking University Health Science Center for rescuing help on XRD analysis. Prof Wenxiong Zhang and Prof Nengdong Wang from Peking University also supported the research by primary XRD analysis.

## Author contributions

Y.-X.H. and Y.-L.J. contributed equally to this work. Y.-X.H., Y.-L.J., Y.Li, J.-H.C. and Z.Y. conceived the project and analysed the experimental results. Y.-X.H., Y.-L.J., Y.Li, H.-X.Y., B.-Q.T. and Z.N. performed the synthesis and characterization. S.L. and Y.Lan performed the theoretical calculations. Y.Lan and Z.Y. composed the manuscript with input from all the authors.

## Additional information

**Competing financial interests:** The authors declare no competing financial interests.

**Publisher's note**: 

