## [Peer review file · Nature Communications]

Reviewers' comments:

Reviewer #1 (Remarks to the Author):

This manuscript describes asymmetric total synthesis of 19-dehydroxyl arisandilactone A. This study features several important contributions including the development of tandem Mukaiyama aldol/oxo-homo-Michael reaction followed by retro-Michael/Michael reactions to establish the core of arisandilactone A. Importantly, the outcome of these studies is well supported by the X-ray crystallography, and detailed understanding of the key steps was accomplished by computational studies.

The only significant drawback is that the authors synthesized 19-dehydroxyl arisandilactone A, and not the actual compound that is found in nature (i.e. arisandilactone A). One may argue that the lack of 19-hydroxy group represents a subtle difference, and that the complexity of this natural product as well as uncovered biosynthetic route justifies publishing this work in Nature Communications. At the same time, one may also argue that the described route (including the biosynthetic proposal) is not going to work for the synthesis of the actual natural compound. Therefore, I would like to request the following before recommending this manuscript for the publication:

- 1) Clearly state in the manuscript that 19-dehydroxyl arisandilactone A is not a natural compound and modify Figure 1 correspondingly (currently it misleads the reader as it states "Naturally occurring norriterpenoids" and includes the structure of 1a).
- 2) If the synthesis is "biomimetically inspired" the authors should include a scheme or figure outlining the biomimetic proposal.
- 3) The computational studies should be applied to the system with the substrates having C19 OMe group to demonstrate that their conclusions actually support biomimetic proposal.
- 4) The SI should include the 2D NMR spectra of the final compound 1a and key reactive intermediates
- 5) The SI should include a table with comparison of the NMR shifts for the synthetic (1a) and actual arisandilactone A (1).

Reviewer #2 (Remarks to the Author):

The manuscript entitled "Biomimetically Inspired Asymmetric Total Synthesis of (+)-19-Dehydroxyl Arisandilactone A" involves the elegant approach for the first asymmetric synthesis of (+)-19-dehydroxyl arisandilactone A through vinylogous Mukaiyama aldol reaction (VMAR), oxa-homo-Michael type reaction, biomimetic retro-Michael/Michael reaction, ring-closing metathesis (RCM) reaction and Cu-catalysed intramolecular cyclopropanation as important key steps starting from (R)-(-)-carvone consists of 37 steps in its longest linear sequence which is an nice work.

The manuscript may be recommended for publication in Nature communication after some revisions.

1. The author should revise the sentence on page 3, "In 2010, Sun9 isolated (+)

arisandilactone A and Shen10 isolated (+) arisandilactone C.....from *S. arisanensis*, which is found in Taiwan." to "In 2010, Shen isolated (+) arisandilactone A and (+) arisandilactone Cfrom *S. arisanensis*, which is found in Taiwan.9, 10"

2. In references 9 and 10 (p.15), the authors should listed the whole authors including the corresponding author Y.C. Shen.
3. In the beginning, we observe a great vision and detail discussion for the target. But, the references for the inhibitory activity toward tumors and hepatitis were not cited in the References.
4. The authors should expect for more distinct and precise result and discussion concerning the yields and challenging steps.
5. Could the authors propose mechanism and cite application references for the key step Oxa-homo-Michael and retro-Michael/Michael reaction?
6. Some differences and mistakes were found between contents and references

Reviewer #3 (Remarks to the Author):

The manuscript entitled "Biomimetically Inspired Asymmetric Total Synthesis of (+)-19-Dehydroxyl Arisandilactone A. Homo-Michael reaction and Tandem Retro-Michael/Michael Reaction in Total Synthesis" (Authors: Yixin Han, Yan-Long Jiang, Yong Li, Hai-Xin Yu, Bing-Qi Tong, Zhe Niu, Shi-Jie Zhou, Song Liu, Yu Lan, Jia-Hua Chen, Zhen Yang) presents a synthesis and a quantum chemical study of the reaction pathway for an important bioactive compound from the Schisandraceae family - (+)-19-dehydroxyl arisandilactone. The authors emphasize, that no synthesis of this compound has been reported earlier. The ability to synthesize such a complicated but promising bioactive system seems to be indeed very important. Since the suggested synthesis involves several steps, where the mechanism is not obvious and different processes could happen simultaneously, the quantum chemical modeling can substantially support the experimental evidence.

Several questions can nevertheless arise for the computational part of the study. The choice of the quantum chemical method should be justified.

For the geometry optimization the B3LYP functional is chosen, though the energies are obtained with M06 functional. Use of different methods for energies and structures is a popular and often a proper compromise, especially if the method used for the energies is too expensive for more demanding structure calculations (e.g. MP2//DFT combination is popular). M06 and B3LYP are both hybrid DFT functionals of the comparable computational cost. Is there a special reason for changing of the DFT approach after the optimization?

The B3LYP functional neglects the dispersion effects, and the M06 takes them only partly into account. Therefore, I would recommend to apply D3-correction in order to check the influence of the dispersion interactions on the computational results.

The authors declare only M06 DFT approach in the main text of the paper, though the supplementary information contains a lot of additional data, such as another method for the structures, the applied basis sets and solvation model. I would recommend also to describe these important details in the main text shortly.

On the pictures 9, 10 and 11 the title of the y-axis should be $\Delta G(M06)$. It is probably only a minor problem with the text format, but in the file, which I was able to download, the symbol " Δ " is corrupted.

In my opinion this manuscript is suitable for Nature Communications, and I recommend the publication with minor revision.

Reviewer #4 (Remarks to the Author):

The authors have submitted some interesting and novel structures for publication. However, the crystal structures submitted are not ready for publication as some of the CIFs include many errors. All have issues that require attention, which should be addressed and the structures resubmitted. In some cases recollection of the data has been recommended (see comments below).

FOR ALL COMPOUNDS

The supplementary information does not contain any experimental information with regards to the data collection for the crystal data submitted e.g. experimental information, diffractometer information etc, nor are there tables reporting the crystallographic refinement information for the submitted crystal structures. A section containing these details must be included. It is recommended that the authors use the latest version of Shelx (ShelX-20XX) to perform the final refinements. This would provide more information about the refinement (instruction file and HKL listing). Embedded refinement details are now required when submitting CIFs.

Compound 10:

- The completeness of the data is poor, despite the crystallographic symmetry being monoclinic. No reasoning for this is given either in the CIF or the supplementary information. This should be addressed either by collecting more data to achieve at least 98% completeness to 0.8Å or by stating a valid reason for the lack of data in either the supplementary information or the CIF.
- Information regarding the crystal `_exptl_crystal_description` and `_exptl_crystal_colour` both missing from the CIF. This information should be included.

Compound 16:

- Information regarding the crystal `_exptl_crystal_description` and `_exptl_crystal_colour` both missing from the CIF. This information should be included.
- Ratio Observed / Unique Reflections (too) Low (25%). This implies that the data is weak or non-existent and that only 25% of the collected data are statistically 2 sigma above the background noise of the experiment. This is unacceptable. This is not addressed in either the supplementary information or the CIF. Perhaps recollection for a longer exposure or collection of the data at a synchrotron would yield better data.
- Structure contains some questionable bond distances e.g. Small Average Phenyl C-C Distances and a C21 - C22 Sp³ carbon bond of 1.615(8) Å. These should be addressed wither through restraints or through refinement against a better data set.

- Large thermal parameters on main molecule, again indicative of a poor data set or lack of suitable restraints.
- Information regarding cell indexation missing. Check.
- Temperature of cell measurement and data collection reported as 293K. Check.
- Information regarding the crystal also missing e.g. description and colour

Compound 21:

- Much experimental information missing from the CIF (device type, cell measurement information, crystal description/colour, absorption correction information). This information must be included.
- Relatively large residual peak in the Fourier difference map. This could be indicative of accounted for disorder or twinning, an error in the absorption correction or bad reflections in the refinement.
- Temperature of cell measurement and data collection reported as 293K. Check.
- Low completeness of data collection, despite the monoclinic crystallographic symmetry. More reflections should be collected to achieve 97% completeness to 0.83Å when using a CuK α source.

Compound 26:

The authors don't appear to have even attempted to ensure that this CIF is competed to publication standards. This CIF is purely a direct output CIF from Shelx97 without inclusion/combination with other CIFs regarding the experiment.

- Basic crystallographic information is missing from the CIF `_symmetry_cell_setting`
- All experimental information missing from the CIF (device type/method, cell measurement information, crystal description/colour/dimensions, absorption correction information, data collection/reduction software). These MUST be included.
- Absolute parameter shift to su on final refinement much greater than 0.2 (9.308). This indicates the refinement did not converge. Additional refinement steps must be performed to achieve convergence. If convergence is not achieved then this indicates an error within the model or the data.
- Large residual peak in the Fourier difference map. This could be indicative of accounted for disorder or twinning, an error in the absorption correction or bad reflections in the refinement.
- Structure contains solvent accessible voids. These are not accounted for in the supplementary information or CIF. Assignment of additional peaks in the difference map or use of SQUEEZE could be employed to account for this void space.
- Ratio Observed / Unique Reflections (too) Low (46%). This implies that the data is weak or non-existent and that only 46% of the collected data are statistically 2 sigma above the background noise of the experiment. This is unacceptable. This is not addressed in either the supplementary information or the CIF. Perhaps recollection for a longer exposure or collection of the data at a synchrotron would yield better data.
- R1 value is high. This is not accounted for in the supplementary information or CIF. Collection of better data or use of a correct model should be used in the refinement.
- Low completeness of data collection, despite the monoclinic crystallographic symmetry. More reflections should be collected to achieve 98% completeness to 0.8Å when using a MoK α source.
- Temperature of cell measurement and data collection reported as 293K. Check.

Compound 29:

- Information regarding the crystal `_exptl_crystal_description` and `_exptl_crystal_colour` both missing from the CIF. This information should be included.
- Flack parameter >0.5 . This could be indicative of miss assigned chirality and the structure being inverted
- Large variations in thermal parameters for non-solvent atoms. Should be addressed with appropriate restraints

Compound 31:

- O1 refined to 0.285(14) occupancy. Is this partial occupancy justified? Combined with short O163 - O1 D - A distance, is the atom correctly assigned?
- Flack parameter meaningless ($su \gg \text{value}$). This should be revised or addressed in the supplementary information or CIF. Method for determining absolute configuration has not been stated in the CIF
- Information regarding the crystal `_exptl_crystal_description`, `_exptl_crystal_colour` and dimensions missing from the CIF. This information should be included.
- Large thermal parameter discrepancy and C41 large ADP max/min ratio. These should be refined using suitable restraints to generate a realistic model.
- Centre of gravity of residues not within the unit cell. Residues should reside mainly within the unit cell.

Compound 32:

- Flack parameter meaningless ($su \gg \text{value}$). This should be revised or addressed in the supplementary information or CIF.
- Ratio of maximum/minimum residual density could be an indication of missed disorder/twinning or the inclusion of bad reflections in the refinement
- Information regarding the crystal `_exptl_crystal_description` and `_exptl_crystal_colour` both missing from the CIF. This information should be included.

Reviewer 1:

Question 1: Clearly state in the manuscript that 19-dehydroxyl arisandilactone A is not a natural compound and modify Figure 1 correspondingly (currently it misleads the reader as it states "Naturally occurring nortriterpenoids" and includes the structure of 1a).

Answer: We changed the title of Figure 1: **"Figure 1 | Naturally occurring nortriterpenoids."** to **"Figure 1 | Naturally occurring nortriterpenoids (1-6) and 19-dehydroxyl arisandilactone A (1a, a derivative of 1)."** See the Track Change Version at page 3.

Question 2: If the synthesis is "biomimetically inspired" the authors should include a scheme or figure outlining the biomimetic proposal.

Answer: We added a paragraph at page 6, first paragraph, line 3 (see the Track Change Version), **"This strategy has been used by Pyne and co-workers in the total synthesis of *Stemona alkaloid 11(S),12(R)-dihydrostemofoline (C)* from *11(S),12(S)-dihydrostemofoline (A)* through intermediate *B* (Figure 3).¹⁸ The stereochemical outcome of the DBU-initiated ring opening reaction of *A* can be rationalized as occurring through a deprotonation of *A* by DBU at the acidic γ -position of the lactone ring would result in the anionic intermediate *B*, which would lead to *11(S),12(R)-dihydrostemofoline (C)* to avoid the unfavorable steric interaction between the methoxy (C13) and methyl (C10) groups in the intermediate *B* (Figure 3)."** We also add Figure 3 at page 6 to describe the biomimetic chemistry.

Figure 3 | DBU mediated a biomimetic tandem retro-Michael/Michael reaction for the formation of 11(S),12(R)-dihydrostemofoline (C) from 11(S),12(S)-dihydrostemofoline (A) through intermediate B

Question 3: The computational studies should be applied to the system with the substrates having C19 OMe group to demonstrate that their conclusions actually support biomimetic proposal.

Answer: Thanks for this comment. As described in the revised manuscript, the 19-dehydroxyl arisandilactone A, which has C19 hydrogen atom, was synthesized in experimental part. The theoretical calculation thus employed 19-dehydroxyl arisandilactone A as the substrate. Our computational results showed that the functional groups on C19 position would not participate in the hydroboration step, the homo-Michael step, or the retro-oxo Michael reaction. Therefore, using the substrates having C19 OMe group in DFT studies will not affect the conclusion. In addition, from the comparison of NMR spectrum of the natural arisandilactone A with our synthetic 19-dehydroxyl arisandilactone A, the

deviation for the chemical shifts of the ^1H -NMR associated with the natural arisandilactone A and our synthetic 19-dehydroxyl arisandilactone A are quite small, except the proton at C19 (see page 74 in SI), indicating both natural arisandilactone A and our synthetic 19-dehydroxyl arisandilactone A adopt similar conformation, and our biomimetic proposal could be applied to the biomimetic synthesis of natural arisandilactone A.

Question 4: The SI should include the 2D NMR spectra of the final compound 1a and key reactive intermediates.

Answer: We have provided the 2D NMR spectra for the compounds:

- 1) **1a** (see Supporting Information at pages 71-73);
- 2) **9** (see Supporting Information at pages 24-26);
- 3) **10** (see Supporting Information at pages 28-30);
- 4) **20** (see Supporting Information at pages 47-49);
- 5) **13S** (see Supporting Information at pages 53-55);
- 6) **26** (see Supporting Information at pages 61-63).

Question 5: The SI should include a table with comparison of the NMR shifts for the synthetic (1a) and actual arisandilactone A (1).

Answer: We have provided a table for the Comparison of NMR spectrum of natural Arisandilactone A (**1**) with synthetic 19-dehydroxyl Arisandilactone A (**1a**) (see Supporting Information at Page 74).

Reviewer 2:

Question 1. The author should revise the sentence on page 3, "In 2010, Sun9 isolated (+) arisandilactone A and Shen10 isolated (+) arisandilactone C.....from *S. arisanensis*, which is found in Taiwan." to "In 2010, Shen isolated (+) arisandilactone A and (+) arisandilactone Cfrom *S. arisanensis*, which is found in Taiwan.9, 10"

Answer: We made the change accordingly; see the Track Change Version at page 3, last paragraph.

Question 2. In references 9 and 10 (p.15), the authors should list the whole authors including the corresponding author Y.C. Shen.

Answer: According to the guidelines of Nature Commun, when the numbers of authors is over 5, only the first author name to be cited. We therefore don't make the changes as suggested.

Question 3. In the beginning, we observe a great vision and detail discussion for the target. But, the references for the inhibitory activity toward tumors and hepatitis were not cited in the References.

Answer: We added a sentence at end of our manuscript: *The biological investigation of the synthesized (+)-19-dehydroxyl arisandilactone A (1a), as well as some advanced intermediates in this total synthesis is currently underway in our laboratory.* See the Track Change Version at page 21, first paragraph, line 3.

Question 4. The authors should expect for more distinct and precise result and discussion concerning the yields and challenging steps.

Answer: We appreciate the reviewer's precise comment, and made some revisions in our text for more distinct and precise result and discussion concerning the yields and challenging steps.

Example 1: in the Track Change Version at page 12, second paragraph, line 3: "Exposure of **20** to a solution of *N,O*-dimethylhydroxylamine hydrochloride and isopropylmagnesium chloride in THF at $-20\text{ }^{\circ}\text{C}$ ³⁷ formed a Weinreb amide in 90% yield, which was reacted with methylmagnesium chloride to afford a ketone in 89% yield. This ketone then underwent Peterson olefination³⁸ by reaction with freshly produced (trimethylsilyl)methylmagnesium chloride in the presence of CeCl_3 ³⁹ in THF at $0\text{ }^{\circ}\text{C}$, and the resultant mixture was worked up by the treatment with silica gel to give an olefin, which, without purification, was then subjected to a desilylation with TBAF to give **21** in 82% yield four the two steps."

Example 2: in the Track Change Version at page 14, first paragraph, line 3, the dew description is "Thus, after intensive experimentation, we found that treatment of **24** with $\text{BH}_3\cdot\text{Me}_2\text{S}$ ⁴³ in THF at $0\text{ }^{\circ}\text{C}$, followed by oxidation with a solution of H_2O_2 (30%) in the presence of a weak base ($\text{Na}_2\text{B}_4\text{O}_7$)⁴⁴, product **26** could be obtained in 65% yield, together with its diastereoisomer (16% yield)."

Example 3: in the Track Change Version at page 14, second paragraph, 2 line, the new description is "In the event, **27** was coupled with **28** in the presence of $\text{BF}_3\cdot\text{Et}_2\text{O}$ in CH_2Cl_2 at $-78\text{ }^{\circ}\text{C}$, and the resultant mixture was then stirred at $-35\text{ }^{\circ}\text{C}$ for 1 h, followed by workup to give product **29** in 66% yield as a single diastereoisomer (Figure 8).

Example 4: in the Track Change Version at page 116, first paragraph, line 1, the new description is: "We therefore, treated **31** with NaOMe in methanol to invert its stereochemistry at C25, and the resultant product was then subjected to Raney nickel-mediated debenzoylation to give product **32** in 73% yield and 85% yield, respectively. Thus, our total synthesis of 19-dehydroxyl arisandilactone A (**1a**) was eventually achieved in 87% yield by oxidation of **32** with DMP."

Question 5. Could the authors propose mechanism and cite application references for the key step Oxa-homo-Michael and retro-Michael/Michael reaction?

Answer: The mechanism for the Oxa-homo-Michael has been illustrated in the Figure 3, intermediate **B**, and the references are cited as references 14-15. For the retro-Michael/Michael reaction, its mechanism has been illustrated in Figure 3 (see page 6), and the reference is cited as references 18.

Question 5. Some differences and mistakes were found between contents and references.

Answer: We have made some corrections during the revision of our text.

Reviewer 3:

Question 1. The manuscript entitled "Biomimetically Inspired Asymmetric Total Synthesis of (+)-19-Dehydroxyl Arisandilactone A. Homo-Michael reaction and Tandem Retro-Michael/Michael Reaction in Total Synthesis" (Authors: Yixin Han, Yan-Long Jiang, Yong Li, Hai-Xin Yu, Bing-Qi Tong, Zhe Niu, Shi-Jie Zhou, Song Liu, Yu Lan, Jia-Hua Chen,

Zhen Yang) presents a synthesis and a quantum chemical study of the reaction pathway for an important bioactive compound from the Schisandraceae family - (+)-19-dehydroxyl arisandilactone. The authors emphasize, that no synthesis of this compound has been reported earlier. The ability to synthesize such a complicated but promising bioactive system seems to be indeed very important. Since the suggested synthesis involves several steps, where the mechanism is not obvious and different processes could happen simultaneously, the quantum chemical modeling can substantially support the experimental evidence.

Several questions can nevertheless arise for the computational part of the study. The choice of the quantum chemical method should be justified.

For the geometry optimization the B3LYP functional is chosen, though the energies are obtained with M06 functional. Use of different methods for energies and structures is a popular and often a proper compromise, especially if the method used for the energies is too expensive for more demanding structure calculations (e.g. MP2//DFT combination is popular). M06 and B3LYP are both hybrid DFT functionals of the comparable computational cost. Is there a special reason for changing of the DFT approach after the optimization?

Answer: Thanks for the comments. The suggestion of MP2//DFT for calculation has been accepted. The MP2/6-311+G(d) method was employed for solvation single point energy calculation in THF solvent. All structures in Figure 10, 11, and 12 have been recalculated using MP2. However, as MP2 functional is too expensive, solvation single point energy calculation on some structures could not be achieved. Specifically, for some structures in Figure 11 and 12, which have more than 90 atoms, there would be more than 1216 basis functions and 1994 primitive gaussians in MP2 calculation. Corresponding calculation would be terminated because the RWF file generated during calculation process is too large. Fortunately, all the MP2 calculations towards the structures in Figure 10 are successful. The computational results are shown below as Figure S1. Besides, another suggestion about the dispersion effects was also considered. B3LYP-D3(BJ) method with 6-311+G(d) was also used to recalculate the solvation single point energies of structures in Figure 10. As shown in the following free energy profile, the values in parenthesis are free energies obtained by B3LYP-D3(BJ) and corresponding values in square brackets are free energies obtained by MP2. Comparison between these two sets of data suggests that the energy values calculated at MP2/6-311+G(d) level of theory are very close to that obtained by B3LYP-D3(BJ)/6-311+G(d). On another hand, the overall free energy profile obtained by MP2 is higher in comparison with M06 calculated data. However, the tendencies shown in these two sets of data are all the same, and the conclusion would not be changed. Therefore, the M06 functional could give reliable energy information for this work. The Figure S1 was added in Supporting Information for clarity.

With regards to the combination of M06 with B3LYP, this strategy has been widely used in Houk and other chemists' work. Its reliability and validity have been demonstrated. Specifically, B3LYP is known as a popular method for DFT calculation and has been proven to give accurate structural information. While Prof. Truhlar developed M06 functional, which is more expensive than B3LYP, could afford more accurate energy information. Therefore, the combination of M06 with B3LYP could efficiently gain reliable structural information and energy information in DFT calculation.

Figure S1 | Energy profiles for the hydroboration of **24** calculated by B3LYP-D3(BJ)/6-311+G(d) and MP2/6-311+G(d) in THF.

Question 2: The B3LYP functional neglects the dispersion effects, and the M06 takes them only partly into account. Therefore, I would recommend to apply D3-correction in order to check the influence of the dispersion interactions on the computational results.

Answer: This suggestion was also adopted in this work. B3LYP-D3(BJ) method with 6-311+G(d) was used to recalculate the solvation single point energies of structures in Figure 10, 11, and 12. Computational result for Figure 10 was given as Figure S1, and corresponding results for Figure 11 and 12 were summarized in Figure S2 and S3. As shown below, the overall free energy profiles in Figure S2 and S3 is lower than that in Figure 11 and 12, although the free energy profile in Figure S1 is higher than that in Figure 10. More importantly, these data indicate that the tendency shown in B3LYP-D3(BJ) calculated data is very similar to that obtained by M06, and the conclusions would remain the same. Consequently, computational results obtained by the method B3LYP-D3(BJ) is consistent with the data in our manuscript. Figure S2 and S3 are added in the Supporting Information for clarity.

Figure S2 | Energy profiles for the homo-Michael step calculated by MP2/6-311+G(d) in THF.

Figure S3 | Energy profiles for the reversible retro-oxo Michael reaction of 29 calculated by MP2/6-311+G(d) in THF.

Question 3: The authors declare only M06 DFT approach in the main text of the paper, though the supplementary information contains a lot of additional data, such as another method for the structures, the applied basis sets and solvation model. I would recommend also describing these important details in the main text shortly.

Answer: Thanks for the comments. We have revised the manuscript and added the description of basis sets and solvation model in page 11. The revised text is also attached below:

Computational study. Density functional theory (DFT) calculations were carried out using the Gaussian 09 program (Gaussian Inc., Wallingford, CT) to determine the diastereoselectivity for the hydroboration step to stereoselectively form 26 from 24 (see Figure 10). The method B3LYP with 6-31G(d) basis set was used for geometry optimizations in gas phase. M06 functional with larger basis set 6-311+G(d) was employed for solvation single point calculations in THF based on gas-phase stationary points using the continuum solvation model SMD (see SI for details).

Question 4: On the pictures 10, 11 and 12 the title of the y-axis should be $\Delta G(M06)$. It is probably only a minor problem with the text format, but in the file, which I was able to download, the symbol " Δ " is corrupted.

Answer: Our response: Thanks for this kind reminder. We have revised the title of the y-axis in Figure 10, 11 and 12.

Reviewer 4:

Question 1: Compound 10: The completeness of the data is poor, despite the crystallographic symmetry being monoclinic. No reasoning for this is given either in the CIF or the supplementary information. This should be addressed either by collecting more data to achieve at least 98% completeness to 0.8Å or by stating a valid reason for the lack of data in either the supplementary information or the CIF. Information regarding the crystal `_exptl_crystal_description` and `_exptl_crystal_colour` both missing from the CIF. This information should be included.

Answer: The poor quality of the X-ray crystallographic data for compound **10** was the result of the crystal that was slightly moved unexpectedly during the data collection of this compound, and this result is not awarded until far afterwards.

Due to the material of compound **10** was not available, and it is difficult to resynthesize it, we could not cultivate the new crystals for its X-ray crystallographic studies. We therefore used the original crystal data for further conversion after XRD analysis. During the time of our refine the X-ray data of compound **10**, we deleted the last groups of images, as a result, the R factor is greatly improved, and seems to be meaningful. We hope that the current X-ray crystallographic data with the 95.9% completeness is acceptable to support the structure of compound **10**.

In addition, we also added a sentence in our revised text to describe this point. *“and its structure was tentatively confirmed through X-ray crystallography; however, the data were of insufficient quality to allow a definitive determination of the structure.”* (see the Track Change Version at page 5, second paragraph, line 2)

Question 2: Compound 16: Information regarding the crystal `_exptl_crystal_description` and `_exptl_crystal_colour` both missing from the CIF. This information should be included. Ratio Observed / Unique Reflections (too) Low (25%). This implies that the data is weak or non-existent and that only 25% of the collected data are statistically 2 sigma above the background noise of the experiment. This is unacceptable. This is not addressed in either the supplementary information or the CIF. Perhaps recollection for a longer exposure or collection of the data at a synchrotron would yield better data.

Structure contains some questionable bond distances e.g. Small Average Phenyl C-C Distances and a C21 - C22 Sp³ carbon bond of 1.615(8) Å. These should be addressed wither through restraints or through refinement against a better data set.

Large thermal parameters on main molecule, again indicative of a poor data set or lack of suitable restraints.

Information regarding cell indexation missing. Check.

Temperature of cell measurement and data collection reported as 293K. Check.

Information regarding the crystal also missing e.g. description and colour.

Answer: The XRD data for compound **16** have been re-collected as suggested. The CCDC number has been provided in the SI.

Question 3: Compound **21**: Much experimental information missing from the CIF (device type, cell measurement information, crystal description/colour, absorption correction information). This information must be included.

Relatively large residual peak in the Fourier difference map. This could be indicative of accounted for disorder or twinning, an error in the absorption correction or bad reflections in the refinement.

Temperature of cell measurement and data collection reported as 293K. Check.

Low completeness of data collection, despite the monoclinic crystallographic symmetry. More reflections should be collected to achieve 97% completeness to 0.83Å when using a CuK α source.

Answer: The XRD data for compound **21** have been re-collected as suggested. The CCDC number has been provided in the SI.

Question 4: Compound **26**: The authors don't appear to have even attempted to ensure that this CIF is competed to publication standards. This CIF is purely a direct output CIF from Shelx97 without inclusion/combination with other CIFs regarding the experiment.

Basic crystallographic information is missing from the CIF `_symmetry_cell_setting`. All experimental information missing from the CIF (device type/method, cell measurement information, crystal description/colour/dimensions, absorption correction information, data collection/reduction software). These MUST be included.

Absolute parameter shift to su on final refinement much greater than 0.2 (9.308). This indicates the refinement did not converge. Additional refinement steps must be performed to achieve convergence. If convergence is not achieved then this indicates an error within the model or the data.

Large residual peak in the Fourier difference map. This could be indicative of unaccounted for disorder or twinning, an error in the absorption correction or bad reflections in the refinement.

Structure contains solvent accessible voids. These are not accounted for in the supplementary information or CIF. Assignment of additional peaks in the difference map or use of SQUEEZE could be employed to account for this void space.

Ratio Observed / Unique Reflections (too) Low (46%). This implies that the data is weak or non-existent and that only 46% of the collected data are statistically 2 sigma above the background noise of the experiment. This is unacceptable. This is not addressed in either the supplementary information or the CIF. Perhaps recollection for a longer exposure or collection of the data at a synchrotron would yield better data.

R1 value is high. This is not accounted for in the supplementary information or CIF. Collection of better data or use of a correct model should be used in the refinement.

Low completeness of data collection, despite the monoclinic crystallographic symmetry. More reflections should be collected to achieve 98% completeness to 0.8Å when using a MoK α source.

Temperature of cell measurement and data collection reported as 293K. Check.

Answer: The XRD data for compound **26** have been re-collected as suggested. The CCDC number has been provided in the SI.

In data for compound **26**, a B alert was reported by <http://checkcif.iucr.org/>, this was due to manually removed crystallized solvent. The solvent molecule could not be assigned clearly, though the crystal was cultivated from ether-methanol-water. This explanation has also been included in the SI.

Question 5: Compound **29**: Information regarding the crystal `_exptl_crystal_description` and `_exptl_crystal_colour` both missing from the CIF. This information should be included.

Flack parameter >0.5. This could be indicative of miss assigned chirality and the structure being inverted.

Large variations in thermal parameters for non-solvent atoms. Should be addressed with appropriate restraints.

Answer:

The crystal `_exptl_crystal_description` and `_exptl_crystal_colour` have been added.

As the enantiomer could be assigned by reference to an unchanging chiral center from compound **26** in the synthetic procedure, this XRD data was collected using Mo K α as diffraction source, which led to big flack parameter. Mo K α was used instead of Cu K α for the crystal was not big enough, and the time for collecting data was limited. Method for determining absolute configuration has also been included in the CIF.

The data has been examined again, and the large variations in thermal parameters came from gem-dimethyl lactone of B ring. This could be attributed to flexible conformation of 5-membered-ring, and the terminal methyl magnify this variations.

Question 6: Compound **31**: O1 refined to 0.285(14) occupancy. Is this partial occupancy justified? Combined with short O163 - O1 D - A distance, is the atom correctly assigned?

Flack parameter meaningless ($su \gg$ value). This should be revised or addressed in the supplementary information or CIF. Method for determining absolute configuration has not been stated in the CIF.

Information regarding the crystal `_exptl_crystal_description`, `_exptl_crystal_colour` and dimensions missing from the CIF. This information should be included.

Large thermal parameter discrepancy and C41 large ADP max/min ratio. These should be refined using suitable restraints to generate a realistic model.

Centre of gravity of residues not within the unit cell. Residues should reside mainly within the unit cell.

Answer: The No. 1 atom has been carefully considered, and according to the surrounding environments and the number of atoms, this atom could only be assigned to be insufficiently occupied oxygen from water, and the relative hydrogen was not added due to insufficient occupancy. Additionally, the short O163-O1 distance could also be attributed to a hydrogen bond. This explanation has also been included in the SI.

Because of the same reason as compound **29**, this data was also collected using Mo K α . `_chemical_absolute_configuration` has been included in the CIF.

The crystal `_exptl_crystal_description` and `_exptl_crystal_colour` have been added.

The data was refined again using suitable restraints as suggested. It did give a better C41 ADP max/min ratio, which might be acceptable. Besides, the center of gravity of residues has also been adjusted within the unit cell.

Question 7: Compound 32: Flack parameter meaningless ($su \gg$ value). This should be revised or addressed in the supplementary information or CIF.

Ratio of maximum/minimum residual density could be an indication of missed disorder/twinning or the inclusion of bad reflections in the refinement.

Information regarding the crystal `_exptl_crystal_description` and `_exptl_crystal_colour` both missing from the CIF. This information should be included.

Answer: Due to the same reason as compounds **29** and **31**, this data was also collected using Mo K α . `_chemical_absolute_configuration` has been included in the CIF.

The data has been examined carefully again and we have to admit that there is no disorder or twinning. Instead, the unsatisfying ratio of maximum/minimum residual density was due to the poor, but also the best we can get, crystal quantity. Different from the final natural product, this intermediate is not that stable at room temperature.

The crystal `_exptl_crystal_description` and `_exptl_crystal_colour` have been added.

REVIEWERS' COMMENTS:

Reviewer #1 (Remarks to the Author):

The authors have addressed all of my prior concerns, and I would like to recommend this manuscript for the publication.

Reviewer #3 (Remarks to the Author):

The manuscript entitled "Biomimetically Inspired Asymmetric Total Synthesis of (+)-19-Dehydroxyl Arisandilactone A. Homo-Michael reaction and Tandem Retro-Michael/Michael Reaction in Total Synthesis" (Authors: Yixin Han, Yan-Long Jiang, Yong Li, Hai-Xin Yu, Bing-Qi Tong, Zhe Niu, Shi-Jie Zhou, Song Liu, Yu Lan, Jia-Hua Chen, Zhen Yang) presents a synthesis and a quantum chemical study of the reaction pathway for an important bioactive compound from the Schisandraceae family - (+)-19-dehydroxyl arisandilactone.

In the second version of the manuscript the authors present among other modifications also the changes to the computational study.

The authors write: "However, as MP2 functional is too expensive, solvation single point energy calculation on some structures could not be achieved. Specifically, for some structures in Figure 11 and 12, which have more than 90 atoms, there would be more than 1216 basis functions and 1994 primitive gaussians in MP2 calculation."

This sentence is not quite correct. First, the MP2 method is an ab initio post-Hartree-Fock quantum chemical method and not a "functional" as e.g. B3LYP. The authors should not mix these very different theoretical approaches: ab initio MP2 and DFT (e.g. B3LYP, M06 etc.). Moreover, the MP2 single point calculations can be nowadays achieved for even larger systems as considered in this paper, because several methods were developed exactly for that purpose: e.g. RI approximation or the linear scaling methods. [1]

Nevertheless, the additional calculations that the authors have performed, indeed verify the application of B3LYP and M06 approaches for the considered molecular systems. The comparison of different quantum chemical approaches (MP2, DFT and DFT-D3), presented in the new version of the SI, makes this study more solid.

In the pdf file, which I downloaded, the symbol " Δ " in the pictures 10, 11 and 12 is still corrupted as it was in the previous version of the manuscript, though the authors claim, that it was fixed. Please check it.

Overall, the paper can be accepted for publication in the Nature Communications journal.

1) S. A. Maurer, L. Clin and C. Ochsenfeld, "Cholesky-decomposed density MP2 with density fitting: accurate MP2 and double-hybrid DFT energies for large systems", J. Chem. Phys. 140, 22412 (2014).

Reviewer #4 (Remarks to the Author):

The authors have satisfactorily addressed almost all of the issues raised from the initial review.

However, there is an issue with the .cif for compound 26. The authors are quite right to model the disordered and poorly defined solvent molecules using the SQUEEZE algorithm. However, SQUEEZE has not been implemented correctly for compound 26 and the authors have incorrectly attributed the type B alert. This is evidenced from the embedded shelx res file in the CIF by the lack of additional parameter terms after the L.S. instruction and a missing ABIN command.

With shelx-20XX the method of implementing SQUEEZE has changed from older methods where SQUEEZE would output a doctored .hkl file. The SQUEEZE calculations are now performed on the outputted .CIF and .FCF files from the shelx-20XX refinement, producing a .fab file containing the solvent scattering factor contribution information. This is then used by shelx-20XX along with the original unchanged .hkl to refine the model.

This must be corrected and the CIF for compound 26 must be reproduced using the correct implementation of SQUEEZE.

All of the other structure files are now satisfactory and ready for publication.

For Reviewer 1:

The authors have addressed all of my prior concerns, and I would like to recommend this manuscript for the publication.

For Reviewer 3:

Comment 1: In the second version of the manuscript the authors present among other modifications also the changes to the computational study. The authors write: “However, as MP2 functional is too expensive, solvation single point energy calculation on some structures could not be achieved. Specifically, for some structures in Figure 11 and 12, which have more than 90 atoms, there would be more than 1216 basis functions and 1994 primitive gaussians in MP2 calculation.” This sentence is not quite correct. First, the MP2 method is an ab initio post-Hartree-Fock quantum chemical method and not a “functional” as e.g. B3LYP. The authors should not mix these very different theoretical approaches: ab initio MP2 and DFT (e.g. B3LYP, M06 etc.). Moreover, the MP2 single point calculations can be nowadays achieved for even larger systems as considered in this paper, because several methods were developed exactly for that purpose: e.g RI approximation or the linear scaling methods.

Answer: We appreciate the precise comment by the reviewer, which will help us to do the computational chemistry in the future.

Comment 2: Nevertheless, the additional calculations that the authors have performed, indeed verify the application of B3LYP and M06 approaches for the considered molecular systems. The comparison of different quantum chemical approaches (MP2, DFT and DFT-D3), presented in the new version of the SI, makes this study more solid. In the pdf

file, which I downloaded, the symbol " Δ " in the pictures 10,11 and 12 is still corrupted as it was in the previous version of the manuscript, though the authors claim, that it was fixed. Please check it. Overall, the paper can be accepted for publication in the Nature Communications journal.

Answer: Thanks for the reviewer's comments. We have checked the symbol " Δ " in Figure 10, 11, and 12. The manuscript is also transferred to a PDF file to ensure the validity of " Δ ". We confirm that the symbol in our manuscript is correct. The corresponding screenshots of Figure 10, 11, and 12 in PDF file are shown below, which suggest that the symbol " Δ " in PDF file is all right. The corruption of " Δ " in reviewer 3 downloaded PDF file might be caused by the different versions of PDF reader.

Figure 10 | Energy profiles for the hydroboration of 24. The values given by kcal/mol are the relative free energies calculated by M06 method in tetrahydrofuran solvent.

Figure 11 | Energy profiles for the homo-Michael step. The values given by kcal/mol are the relative free energies calculated by M06 method in dichloromethane solvent.

Figure 12 | Energy profiles for the reversible retro-oxo Michael reaction of 29. The values given in kcal/mol are the

For Reviewer 4:

Comment 1: However, there is an issue with the cif for compound 26. The authors are quite right to model the disordered and poorly defined solvent molecules using the SQUEEZE algorithm. However, SQUEEZE has not been implemented correctly for compound 26 and the authors have incorrectly attributed the type B alert. This is evidenced from the embedded shelx res file in the CIF by the lack of additional parameter terms after the L.S. instruction and a missing ABIN command.

Answer: The cif file for compound **26** has been reproduced using new implementation of SQUEEZE.